# Unobtrusive monitoring: Statistical dissemination latency estimation in Bitcoin's peer-to-peer network

**David Mödinger**[1]*, **Jan-Hendrik Lorenz**[2], **Rens W. van der Heijden**[1], **Franz J. Hauck**[1]

**1** Institute of Distributed Systems, Ulm University, Ulm, Germany, **2** Institute of Theoretical Computer Science, Ulm University, Ulm, Germany

* david.moedinger@uni-ulm.de

## Abstract

The cryptocurrency system Bitcoin uses a peer-to-peer network to distribute new transactions to all participants. For risk estimation and usability aspects of Bitcoin applications, it is necessary to know the time required to disseminate a transaction within the network. Unfortunately, this time is not immediately obvious and hard to acquire. Measuring the dissemination latency requires many connections into the Bitcoin network, wasting network resources. Some third parties operate that way and publish large scale measurements. Relying on these measurements introduces a dependency and requires additional trust. This work describes how to unobtrusively acquire reliable estimates of the dissemination latencies for transactions without involving a third party. The dissemination latency is modelled with a log-normal distribution, and we estimate their parameters using a Bayesian model that can be updated dynamically. Our approach provides reliable estimates even when using only eight connections, the minimum connection number used by the default Bitcoin client. We provide an implementation of our approach as well as datasets for modelling and evaluation. Our approach, while slightly underestimating the latency distribution, is largely congruent with observed dissemination latencies.

## 1 Introduction

The increasing popularity of Bitcoin [1] and the underlying blockchain have led to many applications and use cases relying on this technology. With a blockchain, an application relies on a distributed ledger, shared between participants, consisting of transactions along with a consensus protocol to decide which transactions are valid and put into the ledger. Many blockchain applications rely on low latencies, such as an automated teller machine (ATM) [2], file storages [3], and in general, marketplaces [4]. For such low-latency applications, knowledge about the expected time required to disseminate a transaction through most parts of the network has many uses. This includes sensible user feedback, the computation of expected processing time for processes involving one or multiple transactions, or to gauge double-spend risk, i.e.,

**Data Availability Statement:** All collected data files are available via zenodo and can be found via the url https://zenodo.org/record/2547396 (also submitted as data validation link). It is also available as http://doi.org/10.5281/zenodo.

2547396. Processed data examples and scripts
used in the creation of this manuscript are available
on github: https://github.com/vs-uulm/CoinView
the same is true for the software: https://github.
com/vs-uulm/btcmon. The sofware is available
under an open source license, i.e., can be archived
or used by third parties.

**Funding:** The authors received no specific funding
for this work.

**Competing interests:** The authors have declared
that no competing interests exist.

estimate how long it would take to notice a double-spend transaction in zero-confirmation transactions.

Live monitoring data on the dissemination times in Bitcoin is available by third parties in large scale measurements, e.g., by bitnodes (see https://github.com/ayeowch/bitnodes) and by researchers [5, 6]. While an interested user could use the data produced by these third parties, this would introduce a possibly unwanted dependency on them. Users would also need to trust those parties and their provided data to be correct, reliable and up to date. Participants could also apply measurement techniques themselves, but these require a large amount of resources, as connections to the majority of network participants are needed. This approach is thus infeasible for typical network participants. Last but not least, this approach is rather conspicuous and does not scale to a large number of users.

## 1.1 Contribution

We enable live measurements of transaction-dissemination latencies in Bitcoin in an unobtrusive fashion that is accessible to typical network participants, using only eight connections, which is the minimum number of connections in Bitcoin. We achieve this by contributing:

- A model of dissemination-latency behaviour of the Bitcoin network using a lognormal distribution including discussions on alternative models.

- An approach to adapt the parameters of such a lognormal distribution to new observations, e.g. changes in the network, with an unknown shift parameter.

- A tool estimating the parameters of the modelled dissemination-latency distribution using only eight connections to the network.

- Datasets over various timeframes and places, to model Bitcoin-network behaviour as well as to evaluate such models and tools.

While the implementation relies on behaviour specific to Bitcoin, the general approach is not as limited. The isolation and estimation of dissemination latencies can be applied to various broadcast networks and mechanics, e.g., peer-to-peer queries.

## 1.2 Roadmap

The structure of this paper is as follows: Section 2 discusses existing network-latency measurement strategies. In Section 3, we introduce the relevant aspects of Bitcoin and its network behaviour. In Section 4 we provide an overview of our network monitoring solution, which uses only a few connections. Section 5 to Section 8 focus on aspects of the network monitoring and its evaluation. Section 5 details the data collection and the resulting data sets. Section 6 focuses on the interpretation and modelling of the collected data. In Section 7, we describe the process to deduce similar results with much fewer connections by a Bayesian mechanism. Lastly, in Section 8 we show the experimental evaluation results of the Bayesian mechanism based on the collected data. An overview of the main steps is shown in Fig 1.

## 2 Related work

Monitoring network properties, such as latency, is widely applied in a multitude of different network types. In this section, we discuss different approaches to latency measurements in similar network environments.

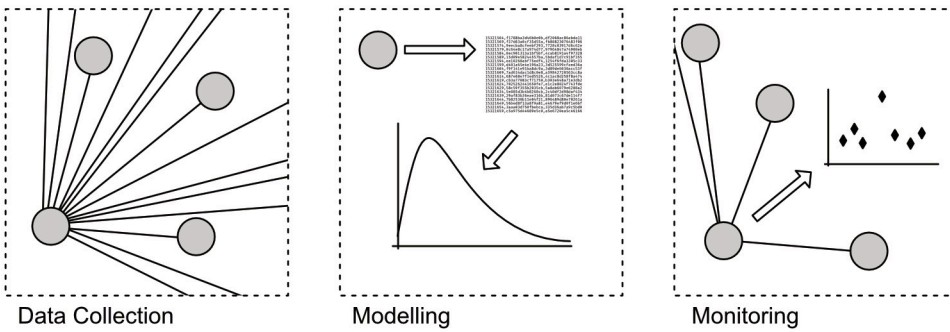

**Fig 1. Overview of the aspects of this paper: Data collection, modelling and monitoring.**

## 2.1 Internet protocol latency measurements

Measurements in general Internet Protocol (IP) networks are a common denominator of network measurements. Although, most IP-level measurements target single-path latencies instead of dissemination latencies.

Yu et al. [7], for example, use an active measurement approach in Software-Defined Networking. They instruct the network devices to route specific control packets through the monitored path. Then they send timestamped packets through the established route. The measured time difference is used to estimate the latency of the targeted path. The estimation is required as there is noise introduced by fluctuations in network behaviour and latency introduced through travelling from and to the in and out routers of the target path. To compute the latency estimates from their measured timestamps they use an estimation distribution, an approach applied on a larger scale in Section 7. The active approach they use still produces strain on a large network and does not scale to the distributed approach required for measuring a broadcast. Breitbart et al. [8] and many others use a similar approach.

Others [9, 10] use passive monitoring, i.e. collecting information on participants and network devices. This requires sharing of collected information to compute a global view on the information, which poses some challenges in a distributed fashion. While this creates reliable data, as long as participants are honest, it also creates additional traffic for information sharing. Shaer et al. [10] use a passive approach to measure latencies and other network properties in IP-multicast environments. They build on the separation of data collection and processing for high-speed analytics. The approaches also require control, or at least dependency and trust, over many network participants to produce reliable results. We attempt to minimise these, due to the negative effects of dependencies and the trust model of Bitcoin. To reduce network overhead, our approach is to infer the desired information from regular traffic instead of actively sharing of measurements.

## 2.2 Peer-to-peer latencies

Bitcoin is built on top of a peer-to-peer network, hence measuring techniques used for general peer-to-peer networks might be applicable. Classical peer-to-peer networks are built on the idea of sharing and, more importantly, finding information. They are not built to spread information to all participants, as Bitcoin does with transactions. Instead, they are built to locate information in a distributed fashion. Locating of information is accomplished through search queries, implemented by flooding techniques or random walks. Therefore, most measurements of classical peer-to-peer networks focus on hop count and search depth of flooding queries instead of dissemination latency [11, 12]. Others, such as Saroiu et al. [13], focused on peer

properties. They only measure pairwise roundtrip latencies between a measurement station and each peer, which are not indicative of in-network latencies over multiple hops and many paths.

Butnaru et al. [14] and Almeida et al. [15] proposed testing and benchmarking frameworks for peer-to-peer networks. These frameworks actively create queries to the network and measure response times. As some of the classical networks, e. g. Gnutella, implement search queries by flooding the network, measurements of the response time for queries is collected by these frameworks. Query response times correlate to dissemination times for rare lookups, but it is imprecise and not considered in these publications.

Active probes are not suitable for the use case of monitoring Bitcoin transaction latencies, as valid transactions can create high costs per probe. They also would require a large number of connections, similar to the fully passive approaches used by third parties in Bitcoin as mentioned earlier.

### 2.3 Bitcoin monitoring

The Bitcoin network has been monitored for various goals. Certain research and private projects [5, 6, 16–18] measure and discuss network properties of Bitcoin. While they actively build connections to participants, they measure the desired traits in a fully passive way: the monitoring software crawls the network for possible clients using the gossip protocol of the network. Then it connects to all found addresses and collects various statistics provided through the network protocol, including user agent, protocol version numbers and more. The software keeps the connection open and logs all received messages with their respective timestamps. This approach allows them to perform accurate measurements through the network. As they track actual traffic instead of probe and control messages, the results are reliable representations of the actual behaviour of nodes.

We apply this approach in Section 6 to collect comparable datasets. We deviate from this for our live monitoring by an abstraction of the desired metric, i.e., latency, and severely reducing the required connections for reliable results.

## 3 Background

This section gives a brief overview of Bitcoin, a blockchain-based cryptocurrency, and the statistical information required for the core results of this paper. As this paper focuses on network behaviour, we restrict the description of blockchains to a high-level understanding.

### 3.1 Bitcoin blockchain

Bitcoin [1] is the first implementation of a so-called blockchain: A distributed data structure of time-stamped transactions between an indeterminate amount of users. To identify these users of the blockchain protocol, Bitcoin uses asymmetric cryptographic keys. Identities are denoted by public keys and possession of the secret key is proven by cryptographic signatures.

The core elements of a blockchain are blocks and transactions. A transaction is an asset transfer. It can have multiple inputs and outputs. An input is a proof of ownership of a previous output, usually a signature proofing the possession of the key used for the output. Inputs can only be used once for a transaction, if two transactions exist referencing the same output, only one can be valid. To decide which transactions are valid, the blocks and mining process is used.

A block is an accumulation of transactions, which also contains a hash of the previous block, forming a chain through these references. The blocks represent the consensus of the system on which transactions are valid. To form a distributed consensus, a mechanism called proof of work is used in Bitcoin and many other blockchains [19]. This proof of work forces

the participants that want to build a block to spend an amount of resources proportional to the available resources in the system. The required resources are usually proportional so that the average rate of block creation is constant. While competing blocks can be created, participants will continue the longest chain, forming a probabilistic consensus on all valid transactions.

In Bitcoin and other permissionless blockchains, everyone can participate in the system. New blocks and transactions need to be transmitted to all participants. As, in principle, everyone can participate, the network protocol is especially important.

## 3.2 Bitcoin network

The underlying network of Bitcoin and, in general, of permissionless blockchains is an unstructured peer-to-peer network. The reference implementation of a Bitcoin client requires a participant to create at least eight connections. Blocks and transactions are broadcast throughout the network [20] by forwarding them through all existing connections. Connected nodes in turn forward to their neighbours.

In principle, this is a flood and prune broadcast: New transactions and blocks are advertised through an inventory message. An inventory message contains identifiers for these new transactions and blocks. A client can then request the actual block or transaction, with a so-called `getdata` message.

To hide the topology of Bitcoin and hide the originator of a block or transaction, the reference implementation does not instantly propagate new information. The Bitcoin-core software creates exponentially distributed values, which are used as waiting times until the next inventory message is sent. See:

1. net_processing.cpp:4140 and

2. net.cpp:2852 of the Bitcoin sources on GitHub, on commit
   `ea595d39f7e782f53cd9403260c6c1d759d4a61a`.

This results in a Poisson point process with an average rate dependent on the expected value of the exponential distribution.

To calculate the waiting times, with an expected average delay of $a$ ms and a minimum of 0.5, Bitcoin uses the formula:

$$\ln(1 - \frac{\text{rand}(0, 2^{48})}{2^{48}})(-10^6)a + 0.5. \tag{1}$$

The recent version `0.20.0` uses a value resulting in an average of 5 seconds as default. To prevent unacceptable long waiting times, Bitcoin caps the generated values at 7 times the average, i.e., 35 seconds. According to the sources, the privacy consideration of outbound connections are different from inbound connections and outbound connections have therefore half the delay.

There are alternative implementations of Bitcoin (Cf. Bitnodes list of user agents) which do not have to follow this implementation. Further, there are proposals for different privacy approaches [21, 22] which are not implemented yet and are therefore not considered during this paper. If they were to be implemented, the strategies of this paper would need adaption and reevaluation.

## 3.3 Time measurements

Given some source node $s$ and some target node $t$ and a measurement node $m$ connected to both. The measurement node receives timestamps $T_s$, $T_t$ by node $s$ and $t$. The difference $T_t - T_s =: M_{s,t}$ is a measurement of some property of the network connecting these two nodes.

First, let us denounce the latency of $n$ successive connections as the term $\ell(n)$. Secondly, the random variable modelling the slowdown of the connection between nodes $i, j$ shall be named $X_{i,j}$. $M_{s,t}$ is then the minimum time taken through the network from $s$ to $t$ through all possible paths between them. Considering the measuring connection slowdown and latency, the result is:

$$M_{s,t} = \min(\text{path}_{s,t}) + X_{t,m} - X_{s,m} + \epsilon(2, \ell). \tag{2}$$

To describe a possible path of length $n$ between nodes, let us denote $\text{path}_n$. Here, $X$ and $X_k$ denote the exponentially distributed random variables between two nodes within the Bitcoin network, without the addition of 0.5. $X_k$ is either Exponentially distributed with $\text{Exp}(\lambda)$ or $\text{Exp}\left(\frac{\lambda}{2}\right)$.

$$\text{path}_1 \quad = \frac{1}{2} + X + \ell(1) \tag{3}$$

$$\text{path}_n \quad = \sum_{k=1}^{n}\left(\frac{1}{2} + X_k + \ell(1)\right) = \frac{n}{2} + \sum_{k=1}^{n} X_k + \ell(n) \tag{4}$$

$$= \frac{n}{2} + \underbrace{\sum_{k=1}^{n-m} X_k}_{\sim \text{Exp}(\lambda)} + \underbrace{\sum_{k=1}^{m} X_k}_{\sim \text{Exp}\left(\frac{\lambda}{2}\right)} + \ell(n) \tag{5}$$

$$= \frac{n}{2} + \underbrace{\sum_{k=1}^{n-m} X_k}_{\sim \text{Erlang}(n-m,\lambda)} + \underbrace{\sum_{k=1}^{m} X_k}_{\sim \text{Erlang}\left(m,\frac{\lambda}{2}\right)} + \ell(n) \tag{6}$$

So a measurement $M_{s,t}$ is a sum of two Erlang, or Gamma, distributions, with some noise linear in the number of participants. Further simplifications of the description of a single path will complicate the notation, as the sum of two Erlang distributions with different scale has no named or well-researched form. Therefore, there is no well-understood model of a minimum of such a sum either. Lastly, with an expected value of 5 seconds, the slowdowns and the $\frac{n}{2}$ term dominate all usual models for latencies of connections and therefore $\ell(n + 2)$.

Given these circumstances, we are interested in the expected time required to reach a given fraction of the network.

## 3.4 Lognormal distribution

We use the lognormal distribution to model behaviour in a later part of this paper, for the discussion on why we use this model, see Section 6. For an in-depth discussion of other distributions used in this paper, we refer to [23].

The lognormal distribution is strongly related to the normal distribution. The normal distribution, also known as Gaussian distribution, is well known for its bell-shaped probability density function. The distribution is defined using two parameters: $\mu$, the mean (and median in case of the normal distribution), and $\sigma^2$, the variance.

The lognormal distribution is a transformation of the normal distribution: The logarithm of the random variable is normally distributed, e.g., given a normally distributed variable $X$, then $e^X$ follows a lognormal distribution. The parameters of a lognormal distribution are

usually given as $\mu$ and $\sigma$ of the underlying normal distribution. Sometimes a third parameter is used, $\gamma$ which represents a shift of the distribution. Parameter estimation of such a shifted lognormal distribution is more complex [24] and will be addressed in Section 7.

The distribution parameters can be used to calculate percentiles. These represent how many of the events modelled by the distribution already happened. The cumulative distribution function could be used to calculate the desired percentiles.

For the normal and lognormal distribution, the cumulative distribution functions are not analytically solvable but can be efficiently approximated. A simple way to approximate the results by [25], with an error $|\epsilon(x)| \leq 5 \times 10^{-4}$, is:

$$\mathrm{cdf}_{\mathrm{lnN}}(x) \quad = \frac{1}{2}\left[1 + \mathrm{erf}\left(\frac{\ln(x) - \mu}{\sigma}\right)\right] \tag{7}$$

$$\approx 1 - \frac{1}{2 \times \left(1 + \alpha_1 s + \alpha_2 s^2 + \alpha_3 s^3 + \alpha_4 s^4\right)^4}, \tag{8}$$

$$s \quad = \frac{\ln(x) - \mu}{\sigma}. \tag{9}$$

Where the values for $\alpha_i$ are constants given by $\alpha_1 = -0.278393$, $\alpha_2 = 0.230398$, $\alpha_3 = 0.000972$ and $\alpha_4 = 0.078108$.

Methods with higher accuracy, i.e., lower error, are available. Let $p \in [0, 1]$, the percentile $100p$ is then calculated by solving $\mathrm{cdf}_{\mathrm{lnN}}(x) = p$ for $x$. This requires either inverting the cumulative distribution function or a numerical search, which can be computed efficiently. Computing the percentile $p$ yields the expected time to reach the fraction $p$ of the network.

## 4 Unobtrusive live monitoring

We introduce a tool (available online at https://github.com/vs-uulm/btcmon) to monitor expected transaction dissemination times in the Bitcoin network. The tool requires only eight connections to produce reliable estimates of the given network behaviour.

### 4.1 Functionality

Our tool produces an estimate of lognormal parameters $\mu$ and $\sigma$. This distribution represents the current dissemination latencies for transactions in the network and can be used to compute the time required to reach a desired fraction of the network. Discussion on the chosen lognormal distribution and possibly other models can be found in Section 6.

Our tool uses initial values for $\mu$ and $\sigma$ and a default of 8 connections. However, these values can be configured by the user. Estimates are generated based on transactions: Data points are collected for each transaction. In principle, there will be one measurement per transaction and connection, showing when a transaction was broadcast by a certain connected neighbour. The monitoring will ignore any data points above the given connection number, due to the mechanism used to adapt the estimations (cf. Section 7).

Technically, the tool consumes text-based input from the standard input. Each line represents one measurement: A timestamp, a node identifier and a transaction identifier. Both identifiers are SHA-256 encoded hashes, but they are treated as arbitrary strings. An example input is given in Box 1.

Box 1. Listing 1. Example of a single input line for the monitoring tool

1547779473468,

6cf1100aaccec75da23995512fc7c7a5b6e25224f5903af011e78691c03d0455,

a73578820a41aa6180621bcd90af1997c88794b33d8db2f004ee37c3e09b10ec

## 4.2 Interpretation of results

The estimates for lognormal parameters by the monitoring can be used to calculate interesting properties of the transaction dissemination. The cumulative probability of the distribution cdf ($t$), therefore, represents the fraction of the network that was likely reached by a given broadcast before a given time $t$.

Given $\mu = 8.5$ and $\sigma = 1$ as a result, the time to reach 75% of the network can be determined as the 75th percentile of the distribution. For the given values, this would be reached at $t \approx$ 9500ms or 9.5 seconds.

## 4.3 Constraints

A client or library needs to be modified to produce logs in the required format for our tool. One such modification is used in Section 5 using bitcoinj. This library is freely available, and necessary modifications are provided in our code repository. Our modified version does not participate in further distributing received transactions to produce quicker and more accurate results.

As this behaviour can be detected and may be suspicious to other participants, a modified client or library could select its measurement times and otherwise behave normally. We recommend relying on the behaviour of the exponential distribution and measure transactions during long pauses created by high values drawn from the distribution. Measurements of transactions produced by the client itself can be used all the time, by sending it to only a single neighbour.

The tool is tuned on data collected from the Bitcoin network. If it is used on a different network, the assumptions and modelling, e.g., is the lognormal distribution applicable, needs to be redone. These assumptions and modelling are described in the following sections.

## 5 Data collection

In a preliminary step, we collected data on the amount and dissemination of transactions in the Bitcoin network. This data is required to model the behaviour of interest (cf. Section 6), i.e., the transaction dissemination latencies, and to evaluate the newly developed estimation tool (cf. Section 8). Note that recreation of this step is not necessary to run the resulting software, but only preparatory to create and validate models.

### 5.1 Related work

Several projects have measured network effects of Bitcoin. We do not consider work that observed the Bitcoin network to deanonymise clients participating in the network [16–18]. Although they are monitoring the network, their results have a different goal.

Bitnodes provides public live data about the Bitcoin network through an application programming interface (API) and web interface. Bitnodes uses the discovery mechanism of the Bitcoin peer-to-peer network to find new peers and connects to them. Information provided by bitnodes includes version numbers of clients and protocols used, nodes distribution over countries, node counts and more. According to the website, the servers connect from a German datacenter.

Coinscope [6] and Neudecker et al. [5] analysed and measured the Bitcoin network to infer its topology. Coinscope is available as standalone modular software and provides large scale monitoring capacity. The software attempts to connect to any reachable node in the Bitcoin network, similar to bitnodes. However, the topology inference techniques rely on outdated behaviour of the Bitcoin core client.

The DSN research group of the Karlsruher Institute of Technology produces similar live monitoring information (https://dsn.tm.kit.edu/bitcoin/) as the one from bitnodes. The group provides information including churn, versions of protocols and clients, node counts, propagation times and more. The information is provided as graphs and tab separated raw datasets. The nodes used to collect this information are located in Germany, and as noted by them, results may vary depending on location.

To validate the data collected by different groups and analyse it further, we collected and provide our own data set. These datasets were taken at worldwide locations and at different but similar points in time.

## 5.2 Collecting methodology

To connect to the network, we modified the library bitcoinj (https://github.com/bitcoinj/bitcoinj/releases/tag/v0.14.7) in version 0.14.7 to add logging capabilities, without modifying any core behaviour. Our modifications on bitcoinj-core are:

a. Generation of a runtime key as a hash of random bytes, to apply a keyed hash to the identities of network participants.

b. Creation of a logging file.

c. For new inventory messages of transactions, the library logs information as a comma-separated line.

The information logged is structured in the following way:

1. Current timestamp of the java virtual machine (JVM) in milliseconds,

2. a keyed hash of the sender identity,

3. the hash of the transaction.

The generation of the runtime key (a.) and keying of sender identities (2.) is added to provide privacy to the participants of the network so that the data can be published. The anonymisation of network participants is done during data collection. No author had access to personal information.

With this modified library, we implemented an application that uses a connection limit of 5000 and does not broadcast received transactions, as to not influence the behaviour intended to measure. This application was run on the local university servers, as well as on Microsoft Azure virtual computers in three regions: Eastern Unites States, South-East Asia and Southern Great Britain. The collection was run for about ten hours each and was rerun on the same virtual machines on multiple dates.

Box 2. Listing 2. Example of a single input line for the monitoring tool

```
docker run –it —rm –v $(PWD):/usr/src/btccol \

    –w /usr/src/btccol —rm openjdk:8 /bin/bash

javac –cp ./bitcoinj –core –0.14.7–bundled.jar ./research/*.java

java –cp ".:./bitcoinj –core –0.14.7–bundled.jar" research/Main
```

**5.2.1 Reproduction.** The modifications for bitcoinj and the application code are available on GitHub (https://github.com/vs-uulm/CoinView, collection subfolder). Further, we provide a precompiled version of the modified library for ease of reproduction. The collection should be reproducible as long as the network will accept the version of the protocol used by the library.

To collect data using these modifications, download the respective files and switch to the `collection/application` subdirectory. Start data collection within a docker container using the commands of Box 2, which are also documented in the repository.

Collected data will be written to a file of the form: "crawler-dd.mm.yyyy hh.mm.ss.csv" where date and time shortages are replaced by the current date and time. The participation in the network in this form is not prohibited by any terms of use.

## 5.3 Info

We collected nine datasets over multiple dates and locations [26], each between 200–670 million individual points of data. Collections from Microsoft Azure provide much fewer data points, as the virtual machines could not create as many connections as the local university server. A description of all datasets can be found in Table 1. All collected datasets are available online (DOI 10.5281/zenodo.2547396) as compressed archives.

## 6 Modelling

To reduce the number of connections needed, we abstract from the collected data to a statistical model. The model represents the frequencies of measured dissemination latencies. This allows us to compute values of interest, such as expected time to reach 90% of the network, using methods discussed in Section 3.4.

**Table 1. All collected datasets of the Bitcoin network.**

| Server position | Nr. | Date | Data points | Note |
|---|---|---|---|---|
| Ulm, Germany | 1 | 2019-01-17 | 570 million | |
| Azure US East | 1 | 2019-01-24 | 207 million | |
| Azure South-East Asia | 1 | | 207 million | |
| Azure Great Britain South | 1 | | 205 million | |
| Ulm, Germany | 2 | 2019-02-01 | 202 million | |
| Ulm, Germany | 3 | 2019-02-06 | 669 million | Same identity key |
| Azure US East | 2 | | 204 million | |
| Azure South-East Asia | 2 | | 203 million | |
| Azure Great Britain South | 2 | | 204 million | |

## 6.1 Methodology

First, to determine the dissemination time for each transaction, we split the dataset by transaction. To simplify the process, assume that the first logged occurrence of a transaction was produced by the originator of the transaction. This assumption is reasonable on average, as the collected data is from a large fraction of the network: Either the originator or a node very close to the originator is present in the data. As a consequence of this assumption, the data is normalised for each transaction by subtracting the timestamp of the assumed originator, i.e., the first entry for the transaction in the log.

The resulting time series for each transaction was then analysed for fitting distributions by visual analysis. SciPy [27] fits reasonable distributions and produces a visual representation of the data and created fits.

## 6.2 Other tested models

We explored several possible distributions before establishing the lognormal distribution as the most fitting model. The tested distributions include a power-law dependency, an exponential, gamma or generalised Pareto distribution. Those distributions are suitable due to their usage in network modelling and relationships to the Poisson distribution created by the privacy mechanism of Bitcoin.

A power law might be applicable for later percentiles of the datasets. Fig 2 shows one evaluation of a possible power law. A linear segment at the end can imply a power-law dependency.

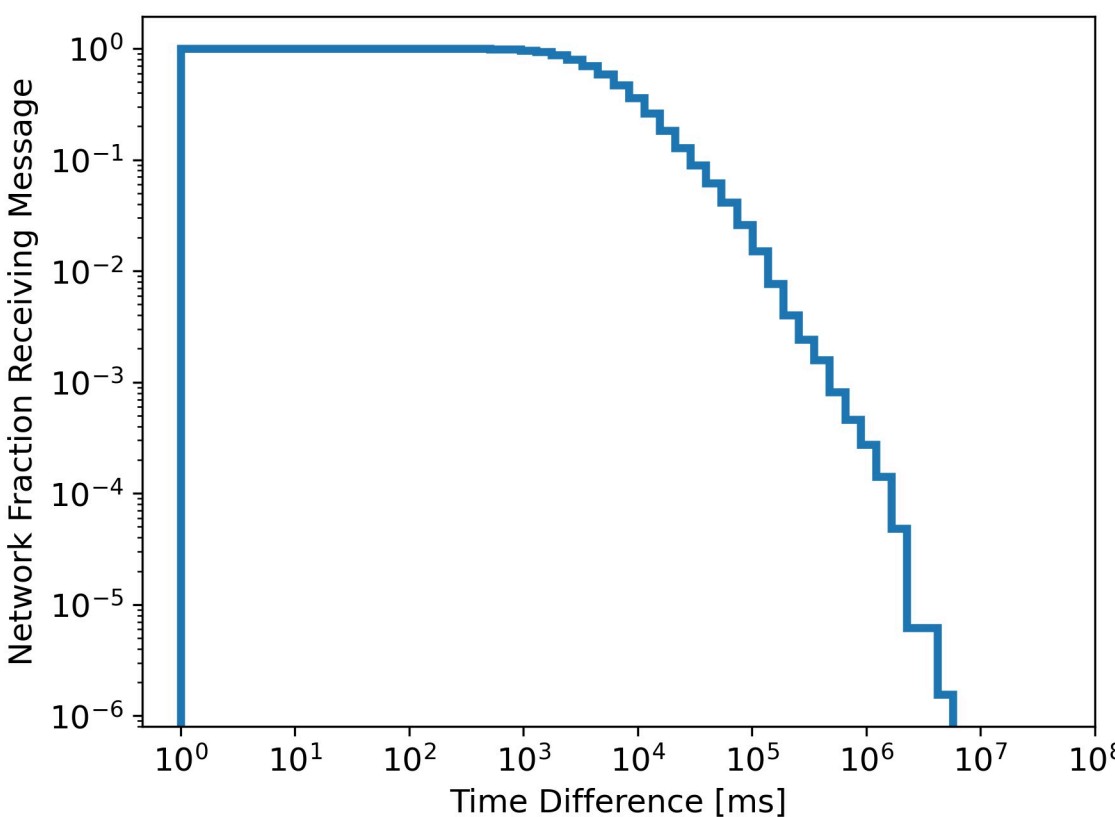

**Fig 2. Density of the data of the February Southern Great Britain dataset in logarithmic scale, a power law dependency should show as a linear dependency in the later part of the visualisation.**

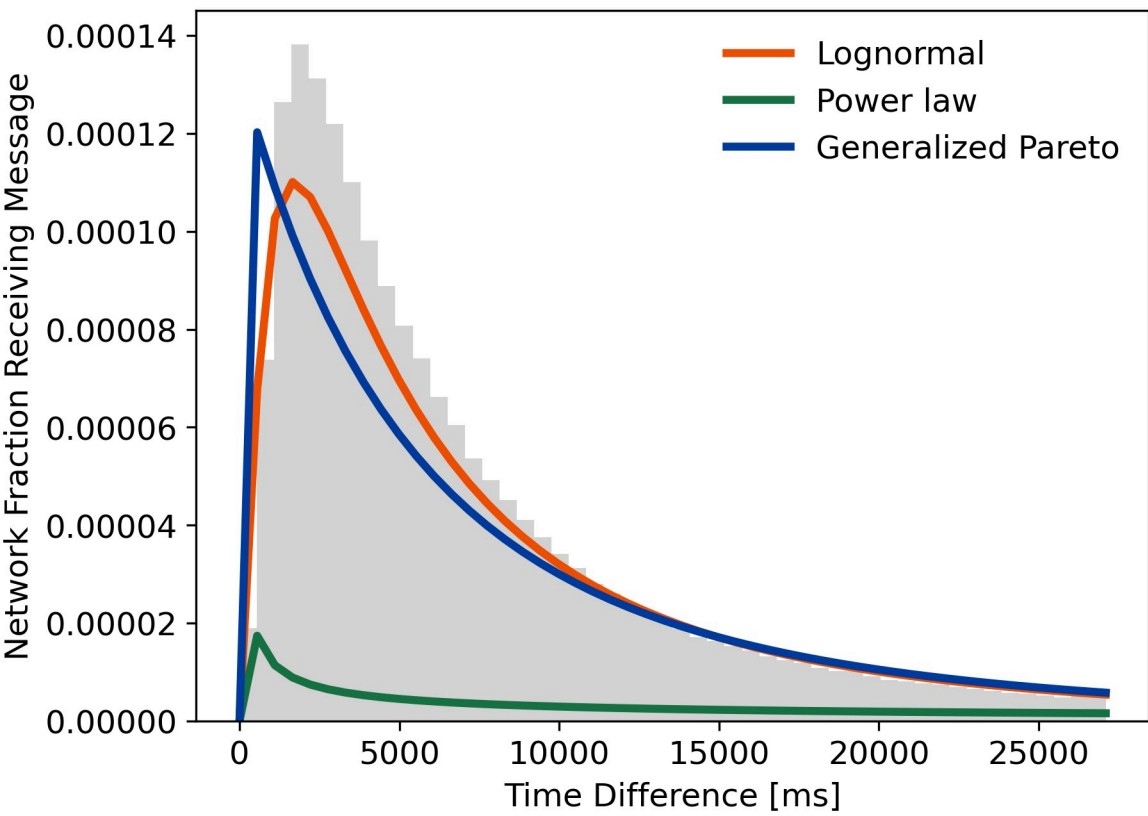

**Fig 3. Evaluations of different distributions for the February Southern Great Britain dataset.** Compares probability density functions and the histogram.

Some of the datasets show a stronger linear end, while most show less of a linear end, implying a power law is not a suitable description of the data.

The gamma and exponential distributions are suitable due to their relation to network modelling and the Bitcoin protocol. Both, and their similar related distributions, do not seem to be a good fit for most of the data. Fig 3 shows an example in the form of probability density functions.

Lastly, the generalised Pareto distribution was chosen due to their competition with the lognormal distribution. Similar to the description in [23], we found the generalised Pareto distribution to describe the extremities better than the lognormal distribution. In contrast, the lognormal distribution is a better fit to describe the main part of the data. If the interest lies more on the tail of the distribution, the generalised Pareto fit will give more accurate results. The generalised Pareto is also included in Fig 3, showing a similar fit as the lognormal distribution.

### 6.3 Lognormal model

While all analysed models produced some outliers, the lognormal distribution described a huge chunk of the data well. Figs 4 and 5 provide normal probability plots of the (base 10) logarithm of several datasets. A normal probability plot shows a linear dependency for normally distributed data, so it should show a linear dependency of the logarithm of the data, for a lognormal distribution. Both, and in general all generated normal probability plots, show a strong deviation of a linear trend in the first and last percentiles. The 95th to 99th percentiles show a medium deviation from the linear trend.

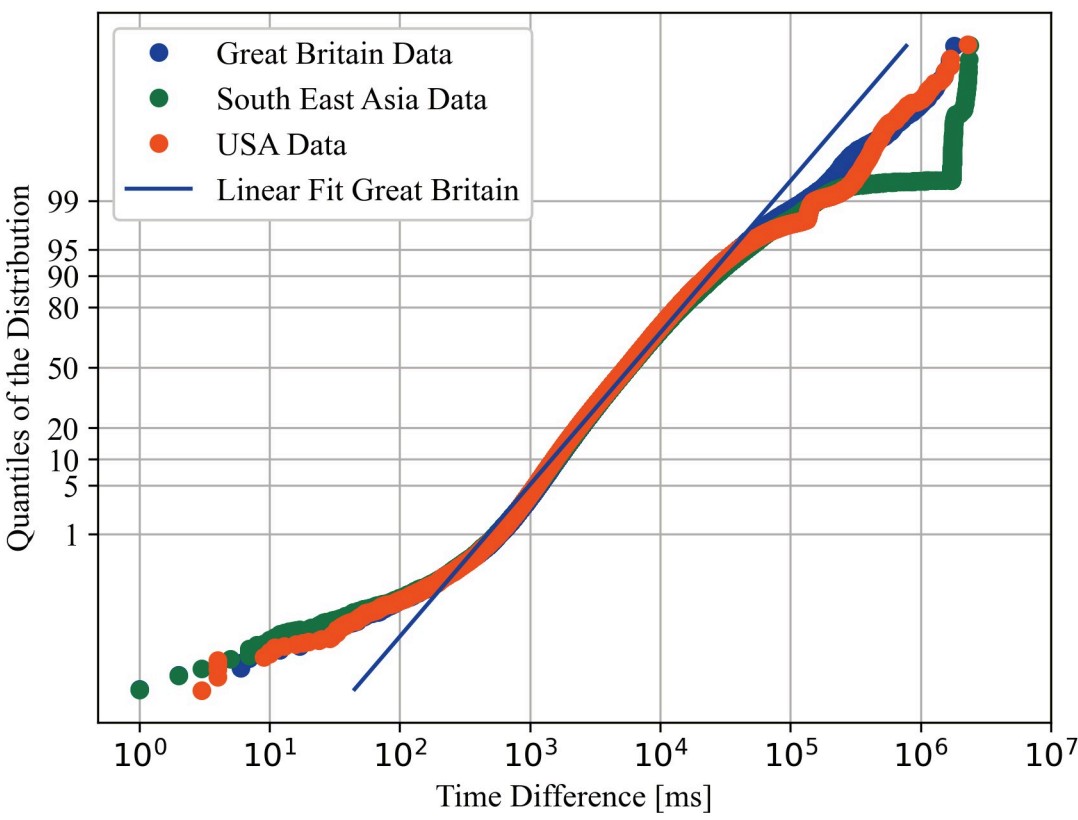

**Fig 4. Lognormal distribution plot of data collected from different Microsoft Azure zones on February 6th.** Data will be reasonably lognormal distributed if it is linear, indicated by a linear fit for the data collected from Great Britain.

Fig 4 shows datasets collected at the same time. As suggested by the DSN Bitcoin monitoring [5], results vary by location, but the variation is small for most parts of the data.

Fig 5 shows datasets produced from the same place at different times. Together with the description in Section 3.3 this strengthens the belief that the model is suitable over time.

One of the nine datasets, the Germany 1 dataset, shows more outliers. These outliers are better explained by a mixed Gaussian distribution over the logarithm of the data. We did not further explore this to reduce the risk of overfitting and due to the lognormal model providing good results of most transactions within this dataset.

## 7 Live adaption of parameter estimates

The live monitoring should not depend on thousands of connections, nor on the collection of huge amounts of data before processing. To solve these problems, we first evaluate schemes to estimate the parameters of a lognormal model containing an unknown shift. A simulation of all approaches, implemented based on the C++ standard library for lognormal random distributions, helps to evaluate the quality of results. Lastly, to improve simulation results, a noise reduction and error compensation scheme is applied.

### 7.1 Parameter inference

As only a low number of connections is desired, we can no longer assume the originator of a transaction is captured in the data. The result is a three-parameter lognormal distribution, with an unknown parameter $\gamma$, which represents the true origination time of the transaction.

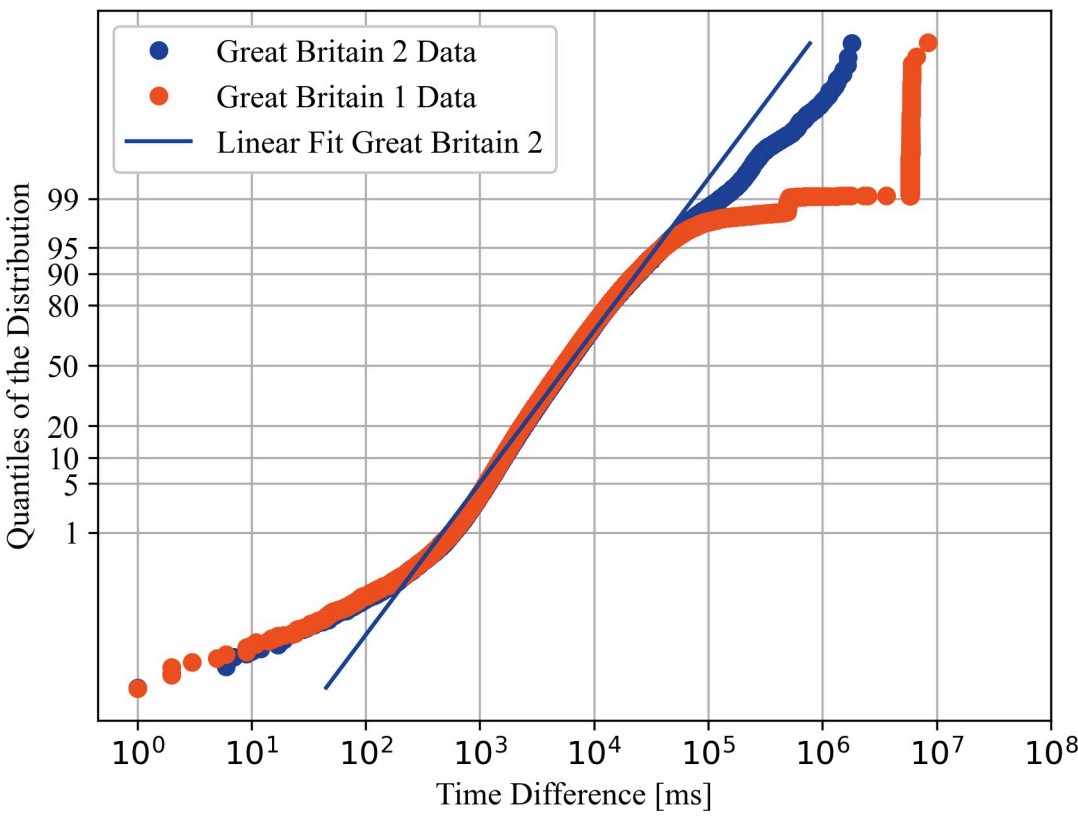

**Fig 5. Lognormal distribution plot of data collected from the Southern Great Britain Microsoft Azure zone on different dates.** Data will be reasonably lognormal distributed if it is linear, indicated by the same fit as in Fig 4.

The data points of a given transaction are considered a measurement of this unknown $\gamma$ shifted distribution.

Based on methods of Iwase and Kanefuji [24] we applied the analytical method, as well as the sampling-based method. The variance and skewness of the measurement $m = \{m_1, \ldots, m_n\}$ a calculated as:

$$\text{skew}(m) \quad = \sum_{i=1}^{n} \frac{(m_i - \text{mean}(m))^3}{n-1}, \tag{10}$$

$$\text{var}(m) \quad = \sum_{i=1}^{n} \frac{(m_i - \text{mean}(m))^2}{n}. \tag{11}$$

This also allows us to calculate $\sigma$ directly.

The analytical method, using the moments skewness and variance result in the following (with $c = \text{skew}(m)$):

$$t \quad = \left( \frac{2 + c^2 + \sqrt{4 \cdot c^2 + c^4}}{2} \right)^{\frac{1}{3}} \tag{12}$$

$$\sigma \quad = \left| \sqrt{\log \frac{t+1}{t-1}} \right| \tag{13}$$

$$\mu \quad = \frac{\log\left(\frac{\text{var}(m)}{\exp(\sigma^2)-1}\right) - \sigma^2}{2} \tag{14}$$

$$\gamma \quad = \text{mean}(m) - \exp(\mu + (\frac{\sigma^2}{2})) \tag{15}$$

The sampling-based approach tries to avoid to directly calculate $\gamma$, as only the parameters $\mu$ and $\sigma$ are of interest. Even further, given a previous estimate, only the difference of the estimated and unknown $\mu$ are of interest, which can be calculated using Algorithm 1.

**Algorithm 1** Algorithm to compute the difference an estimated $\mu$ value and a measurement, without regard for a possible $\gamma$ shift of the source of the measurement.

```
Input: List of measured timesamps m, estimated distribution e, number
of connections c, number of rounds r
Output: Difference of estimated and unknown μ
  m ← {mᵢ - min(m)}
  means ← ∅
  for 1 to r do
    s ← Draw c samples from e
    s ← {sᵢ - min(s)}
    means ← means⋃{mean(s)}
  end for
  return mean(means) - mean(m)
```

We simulated both approaches using a hidden lognormal distribution to generate measurements for the transactions. This can be considered an ideal environment for the algorithm, as the samples obey the distribution without systematic outliers. During the simulation, we noticed both presented approaches produce an error dependent on the parameters of the hidden distribution, and there is a significant amount of noise, due to the low amount of samples used. After some warmup steps, we compared the absolute error as well as the variance of the error. The sampling-based approach produced a mean absolute error of $\approx 0.13$ with a variance of $\approx 0.02$. The formula-based approach produced a mean absolute error of $\approx 17.65$ with a variance of $\approx 0.11$. We focused on the sampling-based approach, as the error and variance of the error is substantially lower.

## 7.2 Bayesian approach

To compensate for the noise of a small number of samples, we apply a Bayesian approach inspired by Kalman filters [28]. Conceptually, the estimates are improved with each measurement, based on the difference between the measurement and the estimate.

Our apriori lognormal estimation is defined by $\mu_e, \sigma_e$. Let $m$ be a measurement of a transaction with eight participants, then the differences of measured and estimated lognormal parameters are:

$$d_\mu \quad = \text{Algorithm1}(m, \ln\text{N}(\mu_e, \sigma_e), 10, 100), \tag{16}$$

$$d_\sigma \quad = \sqrt{\text{var}(m)} - \sigma_e. \tag{17}$$

Given this difference, the estimates are updated based on a fraction of the difference, as there are huge errors in measurements, due to the low number of connections. The update

process is rather simple:

$$\mu_{e+1} \quad = \mu_e + \frac{d_\mu}{c_1}, \tag{18}$$

$$\sigma_{e+1} \quad = \sigma_e + \frac{d_\sigma}{c_2}. \tag{19}$$

For first evaluations $c_1 = 20$ and $c_2 = 2000$ were chosen. Further analysis of the error could improve the speed of convergence, but sufficiently converged values for $\mu$ were reached after the expected 25–30 steps.

Using this adaption mechanism, we detected an additional error between estimations and measurements, even in an ideals world simulation, using lognormal distributions instead of datasets approximating lognormal distributions.

### 7.3 Error compensation

To zero in on the error, we created further simulation experiments. The experiment performed the Bayesian adaption using the sampling-based difference algorithm in Algorithm 1. We restricted the experiment to the modification of $\mu$, i.e., the estimated $\sigma$ was fixed to the $\sigma$ of the hidden distribution. This setup allows us to evaluate the convergence of $\mu$ via the adaption, as more and more measurements are captured.

The resulting error is shown in Fig 6 dependent on the $\mu$ and $\sigma$ values of the hidden distribution. The dependence on $\mu$ is negligible, while the dependence on $\sigma$ is superlinear.

Restricting the analysis to one dimension, i.e. $\sigma$, a simple quadratic fit to the data provides further insights. The results are shown in Fig 7. While this simple error correction mechanism produced good results, we recommend a differently fitted error correction, should the system be applied to networks with huge deviations, outside the highlighted area in the figure.

The fit produced the following error correction function:

$$error(\sigma) = -0.207898 \cdot \sigma^2 + 0.083586 \cdot \sigma - 0.032573. \tag{20}$$

As a result, we added the error correction to the return statement of Algorithm 1.

Absolute Deviation of Lognormal Adaption Parameter μ Dependent on Hidden Parameters μ and sigma

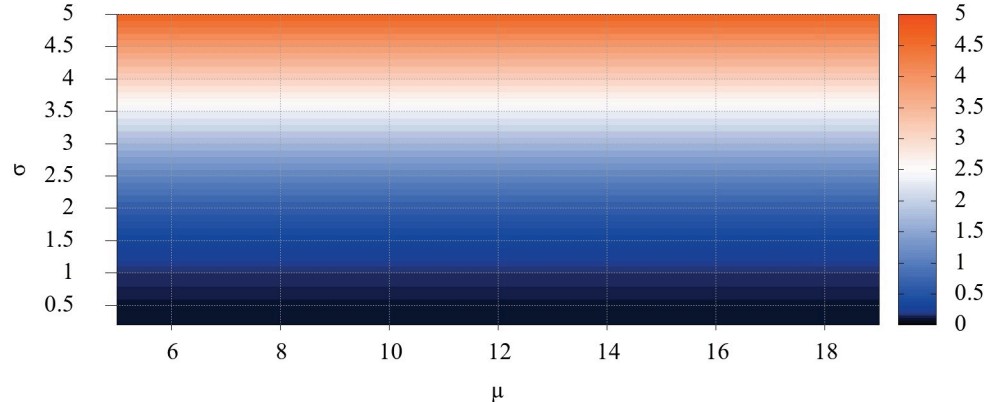

**Fig 6. Heat map of the distance measured between a hidden distribution and our adapted estimates in a simulation.** The x axis shows the dependence of the distance to the $\mu$ parameter, while the y axis shows the dependence on the $\sigma$ parameter. The dependence on $\mu$ is negligible compared to $\sigma$.

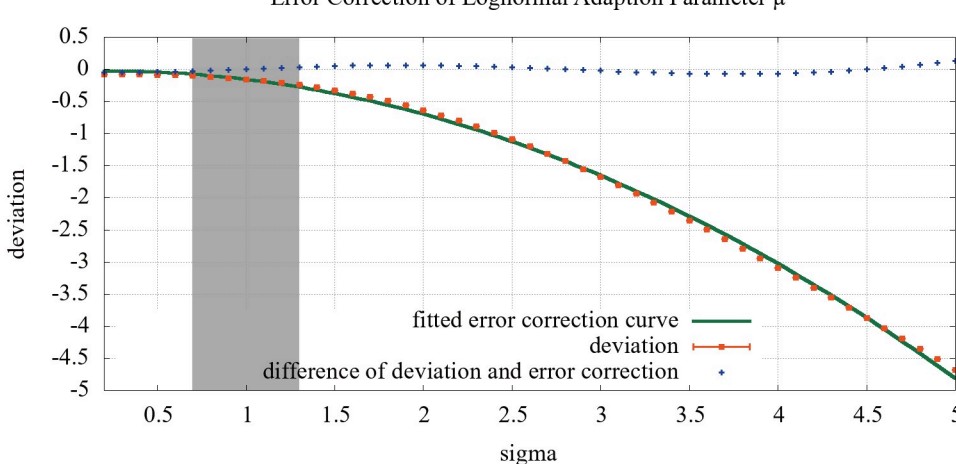

**Fig 7. Simulation results to show the deviation between a hidden distribution and our adapted estimates dependent on $\sigma$, similar to Fig 6, as well as a quadratic fit to correct for the deviation and the difference between the fit and data, i.e., the corrected deviation.**

# 8 Evaluation

This section focuses on the evaluation of the full scheme of monitoring Bitcoin.

## 8.1 Methodology

We use the datasets collected in Section 5 to evaluate the estimations provided by our tool. To reduce evaluation load and remove connection warmup artefacts, we focused on the last 1 million lines of each log for most logs. We also used our largest logs for a long term evaluation by removing the first million lines.

We split each log prepared in this way by participant, i.e., each part contains all log entries received from one network participant. We create one thousand new logs by selecting eight participants at random and merging the logs in chronological order. Each new log represents a log of a virtual node, having connections to only the selected participants.

We ran the monitoring tool on each new log, collecting all estimates over time. As the estimation tool uses initial parameters which require some time to converge, we prepared alternate versions of the results, where the first 30 steps, the warmup phase, has been removed. This was important to have a realistic estimate on the deviation of the results, as some logs might start much later into the original monitoring time, and create a bigger spread by logging values close to the initial values.

The results of all runs were then aggregated into a single result log for each original dataset. The aggregation creates bins of time to collect data. We then calculate the average and standard deviation of all collected data points in each bin.

Per dataset, we calculated the ground truth by splitting the logs by transaction, similar to the method in Section 6.1. Each transaction was then normalised and used to fit a lognormal distribution using SciPy. The fitted parameters were stored with the timestamp of the last contributing log entry, to replicate the process of assigning a time from the estimation. For the ground truth, we did not apply any averaging or further aggregation.

All software used for the evaluation is available in our code repositories (https://github.com/vs-uulm/CoinView and https://github.com/vs-uulm/btcmon).

## 8.2 Results

Fig 8 shows the results of the evaluation of the Great Britain 2 dataset, including warmup steps for the estimation of $\mu$. The adaption of $\mu$ shows a fairly large spread, which can be explained by the original data: The dissemination of transaction is inherently noisy, but the estimates capture the bulk of the data.

Using the estimated parameters $\mu \approx 8.5$ and $\sigma \approx 1.1$ to estimate network behaviour leads to the following latency estimates: The time to reach 50% of all network participants is approximately $e^{8.5} \approx 5000$ms, while reaching 90% would take $\approx 20100$ms.

The long term evaluation using the Germany 3 dataset is shown in Fig 9 for $\mu$ and Fig 10 for $\sigma$. The estimation shows an underestimation of the real-world data for $\mu$ but can capture strong deviations as in the highlighted area. Sparse, long-lasting deviations can not be detected,

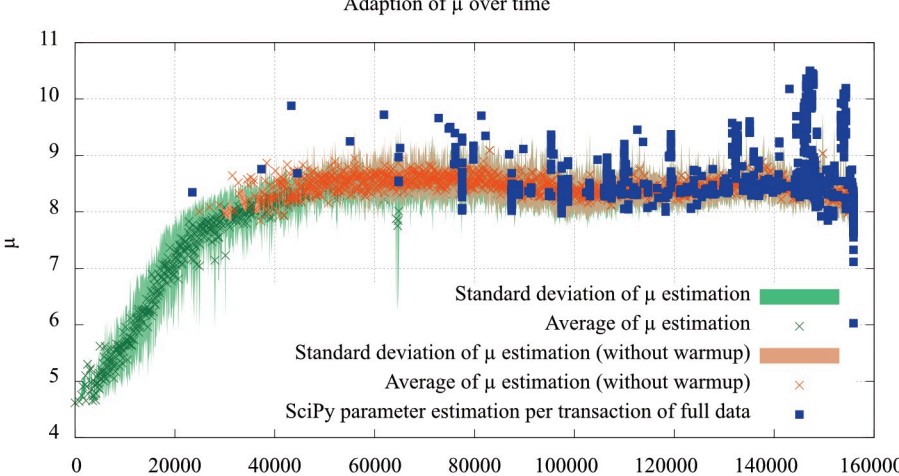

**Fig 8. Evaluation of the $\mu$ parameter estimation using the Great Britain 2 dataset.** The estimates are based on eight randomly selected connections, while the SciPy estimation had access to the full data.

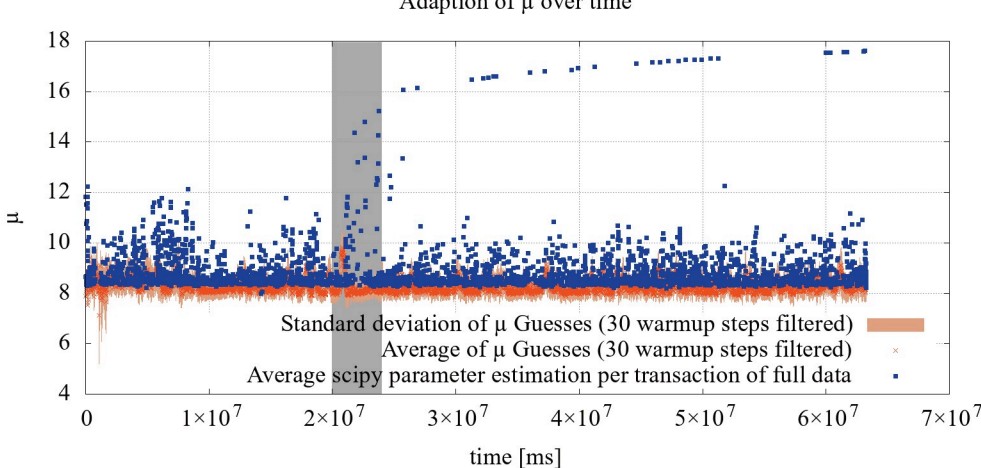

**Fig 9. Evaluation of the $\mu$ parameter estimation using the Germany 3 dataset.** The estimates are based on eight randomly selected connections, while the SciPy estimation had access to the full data.

**Fig 10. Evaluation of the σ parameter estimation using the Germany 3 dataset.** The estimates are based on eight randomly selected connections, while the SciPy estimation had access to the full data.

though. This is expected behaviour, as the estimate attempts to capture overall network performance.

The estimation for σ overestimates the overall network behaviour during this long term test. The reason for this seems to be the conflict of estimation of single transactions versus the overall network behaviour. The results are sufficiently accurate to use them for computations, though. The highlighted area relates to the highlight in the μ adaption, creating a huge spread in the overall data, which can not be captured well by individual transactions.

Overall, the evaluation shows results sufficient for computation of dissemination times in the network. Unfortunately, the real-world data is severely noisy, but the μ estimates capture the bulk of the data well.

## 9 Conclusion

In this paper, we showed that overall dissemination times of transactions in the Bitcoin network can be estimated using the minimum number of connections required by the Bitcoin reference client, i.e., eight connections. Low latency blockchain applications, such as ATMs and file storage applications, profit from such monitoring capabilities for an improved user experience and estimation of double-spend risk.

The monitoring solution is realised by modelling the dissemination times using a lognormal distribution. Such a distribution describes 98% of the collected transaction data well. We provide a proof-of-concept implementation of our monitoring as well as all collected datasets and methodology tools.

The noise created by using a very small number of connections is reduced by a Bayesian scheme to adapt the estimates over several measurements. We also provide a mechanism to determine the difference between the measurement and estimate, circumventing the unknown shift of the real distribution. While the concrete modelling and implementation rely on effects present in Bitcoin, variants are possible for similar networks, and the methodology can be applied to different distributions and models.

The results of the provided tool show good adaption to inherently noisy real-world data independent of geographic location and stable over time.

## Author Contributions

**Conceptualization:** David Mödinger, Jan-Hendrik Lorenz, Rens W. van der Heijden, Franz J. Hauck.

**Data curation:** David Mödinger.

**Formal analysis:** David Mödinger, Jan-Hendrik Lorenz.

**Funding acquisition:** Franz J. Hauck.

**Investigation:** David Mödinger, Jan-Hendrik Lorenz, Rens W. van der Heijden.

**Methodology:** David Mödinger, Jan-Hendrik Lorenz, Rens W. van der Heijden.

**Resources:** David Mödinger.

**Software:** David Mödinger.

**Supervision:** Franz J. Hauck.

**Validation:** David Mödinger, Jan-Hendrik Lorenz.

**Visualization:** David Mödinger, Franz J. Hauck.

**Writing – original draft:** David Mödinger.

**Writing – review & editing:** David Mödinger, Jan-Hendrik Lorenz, Rens W. van der Heijden, Franz J. Hauck.

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
