## [Decision Letter · Decision Letter 0]

27 Oct 2020

PONE-D-20-23629

Unobtrusive Bitcoin network monitoring

PLOS ONE

Dear Dr. Mödinger,

Thank you for submitting your manuscript to PLOS ONE. After careful consideration, we feel that it has merit but does not fully meet PLOS ONE’s publication criteria as it currently stands. Therefore, we invite you to submit a revised version of the manuscript that addresses the points raised during the review process.

We look forward to receiving your revised manuscript.

Kind regards,

He Debiao

Academic Editor

PLOS ONE

Journal Requirements:

Reviewers' comments:

Reviewer's Responses to Questions

**Comments to the Author**

1. Is the manuscript technically sound, and do the data support the conclusions?

Reviewer #1: Partly

Reviewer #2: Yes

2. Has the statistical analysis been performed appropriately and rigorously? 

Reviewer #1: Yes

Reviewer #2: Yes

3. Have the authors made all data underlying the findings in their manuscript fully available?

Reviewer #1: Yes

Reviewer #2: Yes

4. Is the manuscript presented in an intelligible fashion and written in standard English?

Reviewer #1: No

Reviewer #2: Yes

5. Review Comments to the Author

Reviewer #1: The paper discusses existing network-latency measurement strategies and focuses how to unobtrusively acquire reliable estimates of the dissemination latencies for transactions. The dissemination latency is modeled in the Bitcoin network with a log-normal distribution. The paper targets on network behavior and shows that the approach, is largely congruent with actual dissemination latencies. The author explored several possible distributions, before establishing the log normal distribution as the most fitting model. Bayesian scheme is used to adapt the estimates over several measurements. After examining the paper, the following problems were found and the author is suggested to do revisions as mentioned below:

1) The abstract is informative for the purpose and explains the content of the work. But the author is suggested to introduce the problem, identify the main objectives/scope, mention the materials and methods, result and final conclusions.

2) In academic writing, it is important to avoid personal bias. The author is suggested to modify the repeated use of "We" in the article. Using “we” makes the MS about you and your experiences, instead of research and concrete details. Therefore reform such sentences.

3) Acronyms and abbreviations should be clearly defined on their first occurrence in the text by writing the term out in full and following it with the abbreviation in round brackets. For example: Automated teller machine (ATM).

4) Number the equations in the paper.

5) In the conclusion section the author had mentioned "The results could be improved, by further improvements of the error estimation and compensation, as well as the adaption of a full Kalman filter." and “The applicability of other metrics and properties for the process requires a case-by-case investigation, though”.. I suggest the author to provide a synthesis to address the research problem and make a brief summary of the evidence. Finish with some sort of judgment so that the conclusion gets supported by the presented results as discussed.

6) Revise the reference format of the Reference list by including journal name, vol, pp, year.

For example In case of referring to a book/monograph "Rao, C.N.R. & Raveau, Transition Metal Oxides, Wiley, 2nd Edition, 2016, Ch. 2, pp 134-137.

For example in case of referring to a Report "Marine, R.E. & Iliff, K.W. Application of Parameter Estimation to Aircraft Stability and Control. NASA, USA, 1986, Report No. NASA-RP-1168".

7) The quality of images used needs to be improved.

8) In Figures mention the unit of Y axis. If any arbitrary unit is used, use arb. unit, AU, or a.u.

9) Title is the most important element which defines the research study. If it is too short, then it does not tell the reader what is being studied and looks non specific. A good title should provide information about the focus of your research study and utilizes search engine optimization technique to its benefit. Make the title a little long, typically around 8 to 10 words by either highlighting the research problem under investigation or scope of the study or declarative statement or fundamental content.

Reviewer #2: Article is addressing an important issue of cryptocurrency. Risk assessment in bitcoin is very important to gain confidence of the people. Monitoring of bitcoin network is very important to reduce the risk of attack from criminals.

6. PLOS authors have the option to publish the peer review history of their article (what does this mean?). If published, this will include your full peer review and any attached files.

Reviewer #1: **Yes: **Abilash

Reviewer #2: No

---

## [Author Response · Author response to Decision Letter 0]

10 Nov 2020

We thank the reviewers and editors for their generous comments to improve the manuscript. We have edited the manuscript to address their concerns.

In particular we addressed the points raised in the following way:

1) Title changes: We improved the title to make it more specific and provide a better insight into the contents of the manuscript.

2) Abstract improvements: We amended the abstract to give further context for readers.

3) Acronyms: We incorporated the recommended introduction of acronyms on first use.

4) Numbering of equations: We numbered the equations in the manuscript.

5) Conclusion Improvements: We added further context and more precise statements in the conclusion, as recommended by the reviewer.

6) Reference improvements: We added missing information for journals. Unfortunately, not all relevant publications regarding blockchains are academically published. We cleaned up the citations and included information presented in the PLOS ONE reference guide to easily find the referenced works.

7) Overuse of “we”: We reduced the usage of “we” sentences throughout the manuscript as recommended by the reviewer.

8) Additional writing: We corrected some grammar issues and typos.

9) Unlabeled Y Axis: We added labels to the figures missing Y Axis labels. We removed one distribution from one of the graphs, due to errors with the underlying software library. The units of axis are mostly numeric, as common in similar publications in PLOS ONE.

10) Additional images: Reviewed and updated image quality to 300dpi and some smaller improvements.

We believe this addresses the issues raised by the reviewers.

Best regards,

David Mödinger

---

## [Decision Letter · Decision Letter 1]

23 Nov 2020

Unobtrusive monitoring: Statistical dissemination latency estimation in Bitcoin's peer-to-peer network

PONE-D-20-23629R1

Dear Dr. Mödinger,

We’re pleased to inform you that your manuscript has been judged scientifically suitable for publication and will be formally accepted for publication once it meets all outstanding technical requirements.

Kind regards,

He Debiao

Academic Editor

PLOS ONE

Additional Editor Comments (optional):

The author have revised the paper according to reviewers' comments.

Reviewers' comments:

Reviewer's Responses to Questions

**Comments to the Author**

1. If the authors have adequately addressed your comments raised in a previous round of review and you feel that this manuscript is now acceptable for publication, you may indicate that here to bypass the “Comments to the Author” section, enter your conflict of interest statement in the “Confidential to Editor” section, and submit your "Accept" recommendation.

Reviewer #1: All comments have been addressed

Reviewer #2: (No Response)

2. Is the manuscript technically sound, and do the data support the conclusions?

Reviewer #1: Yes

Reviewer #2: (No Response)

3. Has the statistical analysis been performed appropriately and rigorously? 

Reviewer #1: Yes

Reviewer #2: (No Response)

4. Have the authors made all data underlying the findings in their manuscript fully available?

Reviewer #1: Yes

Reviewer #2: (No Response)

5. Is the manuscript presented in an intelligible fashion and written in standard English?

Reviewer #1: Yes

Reviewer #2: (No Response)

6. Review Comments to the Author

Reviewer #1: The aim is almost derivable from the title and synchronizes with that in the abstract and introduction sections. The results were well presented and comprehensible. The paper is now clear, concise, and relevant. The author revised the paper properly, as per the comments given. The manuscript looks free from flaws now.

Reviewer #2: (No Response)

7. PLOS authors have the option to publish the peer review history of their article (what does this mean?). If published, this will include your full peer review and any attached files.

Reviewer #1: **Yes: **Dr. Abilash

Reviewer #2: No

---

## [Editor Report · Acceptance letter]

25 Nov 2020

PONE-D-20-23629R1 

Unobtrusive monitoring: Statistical dissemination latency estimation in Bitcoin’s peer-to-peer network  

Dear Dr. Mödinger:

I'm pleased to inform you that your manuscript has been deemed suitable for publication in PLOS ONE. Congratulations! Your manuscript is now with our production department. 

Kind regards, 

on behalf of

Dr. He Debiao 

Academic Editor

PLOS ONE